# Pre-Trained Model Reusability Evaluation for Small-Data Transfer Learning

**Yao-Xiang Ding**
State Key Lab for CAD&CG
Zhejiang University
yxding@zju.edu.cn

**Xi-Zhu Wu**
National Key Lab for Novel Software Technology
Nanjing University
wuxz@lamda.nju.edu.cn

**Kun Zhou**
State Key Lab for CAD&CG
Zhejiang University
kunzhou@acm.org

**Zhi-Hua Zhou**
National Key Lab for Novel Software Technology
Nanjing University
zhouzh@nju.edu.cn

## Abstract

We study *model reusability evaluation* (MRE) for source pre-trained models: evaluating their transfer learning performance to new target tasks. In special, we focus on the setting under which the target training datasets are small, making it difficult to produce reliable MRE scores using them. Under this situation, we propose *synergistic learning* for building the task-model metric, which can be realized by collecting a set of pre-trained models and asking a group of data providers to participate. We provide theoretical guarantees to show that the learned task-model metric distances can serve as trustworthy MRE scores, and propose synergistic learning algorithms and models for general learning tasks. Experiments show that the MRE models learned by synergistic learning can generate significantly more reliable MRE scores than existing approaches for small-data transfer learning.

## 1 Introduction

Reusing pre-trained models have played essential roles in modern learning pipelines for decreasing training cost, alleviating the requirement of big datasets, and reducing the danger of catastrophic forgetting. The growing number of pre-trained models promotes the birth of large pre-trained model zoos, making it closer towards the future learnware market [Zhou and Tan, 2022]. When selecting a model from these model zoos for doing model transfer, one has to do model reusability evaluation (MRE) first: evaluating the transfer learning performance of the models to the target task and identifying the best model. The role of MRE is crucial since no matter how good the transfer learning strategy is, incorrect MRE would still lead to the danger of negative transfer. MRE has received growing attention in recent years [Achille et al., 2019, Tran et al., 2019, Nguyen et al., 2020, Wu et al., 2020, Ding and Zhou, 2020, You et al., 2021]. But most existing studies focus on large-data MRE, under which the target dataset is sufficiently large.

In this work, we focus on small-data MRE, under which the target training datasets are small. Reusing pre-trained models is essential under the small-data scenario since learning from scratch is difficult. Unfortunately, large-data MRE approaches are usually invalid for small data since they usually focus more on simplicity and efficiency, but not generalization and robustness, which are essential for small-data MRE. It is indeed challenging to obtain reliable MRE results under the small-data scenario due to the fundamental burden set by the laws of statistics.

36th Conference on Neural Information Processing Systems (NeurIPS 2022).

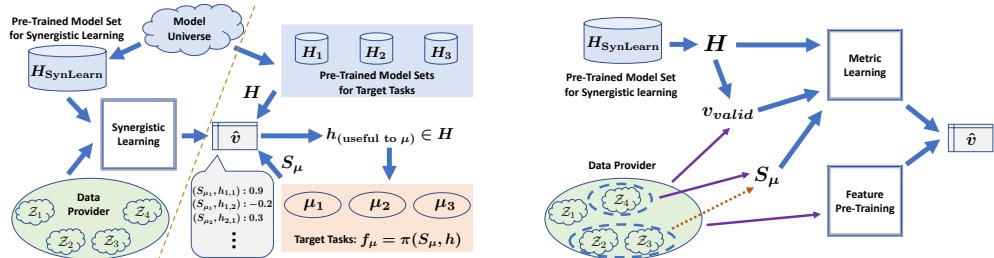

Figure 1: The illustration of the proposed approach. The left figure illustrates the overall procedure, in which the left part illustrates the synergistic learning stage and the right part illustrate the testing MRE stage. The right figure illustrates Algorithm 1. The arrows started from the data providers denote the queries, in which the solid purple ones are SQUERY and the dashed orange one is MQUERY.

Our solution is based on two observations. First, model reusability usually depends on the task-model relationship other than the task-independent property of any specific small sample. We could grasp this relationship by meta-learning an MRE function among tasks and models. Second, this learning process is often realizable in practice, in special for large model zoo platforms where sufficient pre-trained models and data providers are available. Based on these observations, we conduct the following studies in this paper:

**Problem formuation.** We provide the formulation of small-data MRE and synergistic learning to learn the MRE function by metric learning. Synergisitc learning works for general learning scenarios beyond classification, on which most previous MRE approaches focus.

**Theoretical analysis.** We propose access risk analysis showing that the MRE model learned by synergistic learning guarantees to generate reliable MRE scores. The theory not only provides guarantee even for using non-convex deep networks as the MRE model, but also motivates feature pre-training in synergistic learning.

**Algorithm and model design.** We propose synergistic learning algorithms and MRE model structures for general learning tasks, which have the auxiliary advantage of protecting data privacy. Meanwhile, we propose a more elaborate MRE model strucsture for classification.

**Experimental verification.** Experimental results show that synergistic learning can generate significantly more reliable MRE scores for small-data transfer learning than existing MRE approaches.

## 2 Problem Setup

In this section, we provide the formal definition of small-data MRE and synergistic learning.

### 2.1 Small-Data MRE

Denote by $\mathcal{Z} = \mathcal{X} \times \mathcal{Y}$ the observation space, in which $\mathcal{X}, \mathcal{Y}$ are the input and output spaces respectively. A target learning task $\mu$ can be represented by a probability measure[1] $\mu(x, y)$ over some $\mathcal{Z}_\mu \subseteq \mathcal{Z}$. Furthermore, we assume that all $\mu$ are drawn from the task environment $\mathcal{T}$, a probability measure over the space of all target tasks.

In small-data MRE, target tasks $\mu$ are drawn from $\mathcal{T}$. For each $\mu$, the objective is to learn a prediction rule $f_\mu$ such that for any observation $(x, y) \sim \mu$, $f_\mu(x)$ is close to $y$. The learner of $\mu$ will be given a target training dataset $S_\mu$, a set of pre-trained models $H$, a model transfer strategy $\pi$ and an MRE function $v_\pi$. $S_\mu = \{z_k = (x_k, y_k)\}_{k=1}^K$ is a sample drawn from $\mu$ such that the sample size $K$ is a small number. $H$ is the pre-trained model set in which each model $h \in H$ is drawn from the model environment $\mathcal{M}$, a probability measure over the space of all pre-trained models. We assume that the models in $H$ are sampled independently from $\mu$ while different $H$ can be given for different $\mu$. $\pi(S_\mu, h)$ is a transfer learning strategy which can be used to transfer a pre-trained model $h$ into $f_\mu$ using $S_\mu$. We do not restrict the choice of $\pi$, but assume that $\pi$ is fixed for any target task. The MRE function $v_\pi(S_\mu, h)$ is a real-valued function taking $S_\mu$ and $h$ as the inputs

---

[1]We will use $\mu$ to denote both a task and its probability measure below when there is no ambiguity.

and outputting the transferability score. Below we use $v$ instead of $v_\pi$ to simplify notations. Without loss of generality, among all possible $v$, we define $v^*$ as the optimal MRE function such that for any $\mu, S_\mu, h_1, h_2, v^*(S_\mu, h_1) < v^*(S_\mu, h_2)$ if and only if $h_1$ has better transfer learning performance with $S_\mu$ than $h_2$. Thus we assume that the MRE score is *monotonically decreasing* w.r.t. transfer performance. If no special properties exist for $v^*$, estimating $v^*$ would be extremely difficult for small-data MRE. Fortunately, in general, we could assume that model reusability usually depends on the task-model relationship other than the task-independent property of any specific small sample, as discussed in Section 1. Therefore, we assume that *the ground-truth model transferability depends only on the target task $\mu$ but does not depend on the random draw of $S_\mu$ from $\mu$*, i.e. $\forall S_\mu, S'_\mu \sim \mu, v^*(S_\mu) = v^*(S'_\mu)$. Thus we can use the notation $v^*(\mu, h)$ instead of $v^*(S_\mu, h)$ for any $S_\mu, h$. How this assumption can be relaxed is discussed in Section 7.

## 2.2  Synergistic Learning

Synergistic learning is the preparation stage for learning an MRE function $\hat{v}$ that accurately approximate $v^*$ before any MRE problem defined in Section 2.1 starts. A synergistic learning process works with the following three prerequisites:

- A set of pre-trained models $H_{\text{SynLearn}} = \{h_k\}_{k=1}^{N_m}$ drawn from $\mathcal{M}$ are given. Note that the models in this set could be different from models given in any future MRE problems;

- A set of data providers $\mathcal{D} = \{\mathcal{Z}_d\}_{d=1}^{D}$ participate in learning, such that each data provider represents an observation subspace $\mathcal{Z}_d \subseteq \mathcal{Z}$. Furthermore, $\cup \mathcal{Z}_d = \mathcal{Z}$. We define the closure of any task $\mu \sim \mathcal{T}$ as $\bar{\mathcal{Z}}_\mu = \min_L \{\mathcal{Z}_{d_1}, \mathcal{Z}_{d_2}, \dots \mathcal{Z}_{d_L}\}$ s.t. $\mathcal{Z}_\mu \subseteq \cup_{l=1}^{L} \mathcal{Z}_{d_l}$.

- A validation MRE function $v_{valid}$ is provided for synergistic learning. $v_{valid}$ is unbiased between any task and model[2]: $\forall \mu, h, v_{valid}(\mu, h) = \mathbb{E}_{S_\mu \sim \mu}[v_{valid}(S_\mu, h)] = v^*(\mu, h)$.

*The basic idea of synergistic learning is to establish a metric $\hat{v}(S_\mu, h)$ between the target training datasets and the models so that the metric distance could be used as the MRE score.* In Section 4, we will show that learning $\hat{v}$ only requires the data providers to answer two kinds of queries:

- SQUERY$(\psi_s, \mathcal{Z}_d, \mu, M)$: a *single* data provider $\mathcal{Z}_d$ is queried to sample $M$ observations from a given probability measure $\mu$ over $\mathcal{Z}_d$, and return the output of $\psi_s(\{z_i\}_{i=1}^{M})$, a function of the observations.

- MQUERY$(\psi_m, \{\mathcal{Z}_{d_i}\}_{i=1}^{Q}, \{\mu_i\}_{i=1}^{Q}, \{M_i\}_{i=1}^{Q})$: *multiple* data providers $\{\mathcal{Z}_{d_i}\}_{i=1}^{Q}$ are queried jointly. Each data provider $\mathcal{Z}_{d_i}$ sample $M_i$ observations from a given probability measure $\mu_i$ over $\mathcal{Z}_{d_i}$. Aggregating the sampled observations from all data providers, the output of $\psi_m(\{z_{1,j}\}_{j=1}^{M_1}, \{z_{2,j}\}_{j=1}^{M_2}, \dots, \{z_{Q,j}\}_{j=1}^{M_Q})$, a function of all observations, is returned. Furthermore, $\psi_m$ cannot be realized by aggregating multiple SQUERY.

The main difference between these two queries lies in the scope of involved data providers, which is important when the data privacy is sensitive. SQUERY is answered by a single data provider. How this kind of queries can be answered in privacy-guaranteed ways has received many studies [Zinkevich et al., 2010, Konečný et al., 2016]. In contrast, MQUERY needs to be answered by multiple data providers jointly. Protecting data privacy is much harder in this situation since their data need to be aggregated. Therefore, we set an auxiliary target of reducing the number of times to use MQUERY.

Finally, for learning $\hat{v}$, we assume that $\hat{v}$ is formed by three parts: (1) $g_\mu(S_\mu)$, the task feature backbone; (2) $g_h(h)$, the combination of the model specification generator[3] and the model feature backbone; (3) $d_\theta(c_\mu, c_h)$, the metric module, in which $c_\mu$ is the output of $g_\mu(S_\mu)$, $c_h$ is the output of $g_h(h)$, and $\theta$ is the learnable parameter of $d_\theta(c_\mu, c_h)$. $g_\mu$ and $g_h$ transfer target training dataset $S_\mu$ and pre-trained model $h$ into their representations $c_\mu$ and $c_h$. The metric module then calculates the metric distance between $c_\mu$ and $c_h$ as the MRE score. The details are introduced in Section 4.2.

---

[2]Even though $v_{valid}$ is unbiased, it may have high variance when $S_\mu$ is small. Thus $v_{valid}$ can not be used directly for small-data MRE.

[3]Please refer to Section 7 for details about model specifications.

# 3 Theoretical Analysis

In this section, we discuss the theoretical foundation of synergistic learning. Readers who are interested more on algorithmic ideas can skip this section without affecting understanding significantly. In the analysis, we assume that $g_\mu$ and $g_h$ are fixed and focus on learning the metric module. Thus the learnable parameter for $\hat{v}$ is $\theta$.

For metric-based synergistic learning, we assume that $N_m$ pre-trained models $\{h_k\}_{k=1}^{N_m}$ sampled from $\mathcal{M}$, $N_t$ tasks $\{\mu_i\}_{i=1}^{N_t}$ sampled from $\mathcal{T}$ and $N_S$ target training datasets $\{S_{\mu_i,j}\}_{j=1}^{N_S}$ sampled from each task $\mu_i$ are used for training. We also assume that for any pair of $h_i$ and $S_{\mu_j,k}$, the ground-truth MRE score $v^*(\mu_i, h_k)$ are given. Define $r_{v^*}(S_\mu, S_{\mu'}, h, h'; \hat{v})$ as $\mathbf{I}[\Delta_{v^*}(\mu, \mu', h, h') < 0]\mathbf{I}[\Delta_{\hat{v}}(S_\mu, S_{\mu'}, h, h') \geq 0]$ in which $\Delta_{v^*}(\mu, \mu', h, h') = v^*(\mu, h) - v^*(\mu', h')$, $\Delta_{\hat{v}}(S_\mu, S_{\mu'}, h, h') = \hat{v}(S_\mu, h) - \hat{v}(S_{\mu'}, h')$, and $\mathbf{I}$ is the indicator function. Since $v^*$ is the ground-truth MRE function, a desirable $\hat{v}$ should minimize

$$R(\hat{v}) = \mathbb{E}\big[r_{v^*}(S_\mu, S_{\mu'}, h, h'; \hat{v})\big],$$

where the expectation is taken over $S_\mu \sim \mu, S_{\mu'} \sim \mu', \mu, \mu' \sim \mathcal{T}, h, h' \sim \mathcal{M}$. While for the convenience of optimization and analysis, we define our objective using the triplet surrogate loss. Define $r_{v^*}^\gamma(S_\mu, S_{\mu'}, h, h'; \hat{v})$ as $\mathbf{I}[\Delta_{v^*}(\mu, \mu', h, h') < 0][\Delta_{\hat{v}}(S_\mu, S_{\mu'}, h, h') + \gamma]_+$ in which $[x]_+ = \max\{x, 0\}$ and $\gamma > 0$. When $\gamma > 1$, $r_{v^*}^\gamma$ upper bounds $r_{v^*}$. Meanwhile, $r_{v^*}^\gamma$ is consistent with $r_{v^*}$ since $r_{v^*}^\gamma(S_\mu, S_{\mu'}, h, h'; \hat{v}) = 0$ indicates that $r_{v^*}(S_\mu, S_{\mu'}, h, h'; \hat{v}) = 0$. Therefore, we define our goal as minimizing the following expected risk

$$R^\gamma(\hat{v}) = \mathbb{E}\big[r_{v^*}^\gamma(S_\mu, S_{\mu'}, h, h'; \hat{v})\big]. \tag{1}$$

A possible way to achieve this goal is to minimize the following empirical risk

$$\hat{R}^\gamma(\hat{v}) = \frac{1}{(N_t N_S N_m)^2} \sum \big[r_{v^*}^\gamma(S_\mu, S_{\mu'}, h_k, h_{k'}; \hat{v})\big] \tag{2}$$

where the summation is taken over all models, tasks and datasets used for training. We want to know whether minimizing the empirical risk $\hat{R}^\gamma(\hat{v})$ indeed leads to the minimization of the expected risk $R^\gamma(\hat{v})$. Denote by $\Theta$ the parameter space for $\theta$ and use $\hat{v}_\theta$ to denote the $\hat{v}$ with learnable parameter $\theta$. We provide an access risk bound showing the effectiveness of minimizing the above empirical risk. Use $\bar{G}_\theta(S_\mu, S_{\mu'}, h, h')$ to denote $[\Delta_{\hat{v}_\theta}(S_\mu, S_{\mu'}, h, h') + \gamma]_+$. Additionally, we assume that $\nabla[\bar{G}_\theta(S_\mu, S_{\mu'}, h, h')] = 0$ when $\bar{G}_\theta(S_\mu, S_{\mu'}, h, h') = 0$. Furthermore, let

$$\nabla[\hat{R}(\bar{G}_\theta)] = \frac{1}{(N_t N_S N_m)^2} \sum \|\nabla^-[S_\mu, S_{\mu'}, h, h']\|,$$

in which the summation is taken over all tasks, datasets and models used for training and we have $\nabla^-[S_\mu, S_{\mu'}, h, h'] = \mathbf{I}[\Delta_{v^*}(\mu, \mu', h, h') < 0]\nabla[\bar{G}_\theta(S_\mu, S_{\mu'}, h, h')]$. Meanwhile, we denote $V = \max_{i \in [N_t]} \mathbb{E}_{S_{\mu_i} \sim \mu_i}[c^2(S_{\mu_i})]$. Now we are ready to state the following theorem whose proof is provided in the appendix (Section A).

**Theorem 3.1.** *Under Assumption A.1, there exist optimal parameter $\theta^* \in \Theta$ and constant $\beta > 0$, $\forall \theta \in \Theta$, the following event*

$$R^\gamma(\hat{v}_\theta) - R^\gamma(\hat{v}_{\theta^*}) \leq \beta\big[\nabla[\hat{R}(\bar{G}_\theta)] + \Delta(N_t, N_m, N_S)\big]$$

*holds with high probability, in which $\Delta(N_t, N_m, N_S) = \tilde{O}(1/\sqrt{N_t}, 1/\sqrt{N_m}, V/\sqrt{N_S}, 1/N_S)$.*

Theorem 3.1 shows several interesting insights. First, $\nabla[\hat{R}(\bar{G}_\theta)]$ is the gradient norm of the metric distance gaps from the training data. The bound shows that the access risk will be small when the gradient norm tends to zero, even when the global minimum of the empirical risk has not been reached. This makes the bound informative when non-convex models, such as DNNs, are used. Second, the bound has the normal $O(1/\sqrt{N_t}), O(1/\sqrt{N_m})$ sample complexity dependence for both tasks and models. This is in agreement with our intuition such that lacking any of the tasks and models would lead to the failure of learning. Finally, the most interesting take-away is the sample complexity for target datasets in each task, which has a dependence of the feature variance $V$. If $V$ is small, the order becomes $O(1/N_S)$ instead of $O(1/\sqrt{N_S})$, a significant drop for the number of the training datasets. In Section 4, we show that this result inspires us to include feature variance reduction in the synergistic learning process, which would significantly reduce using MQUERY.

---

**Algorithm 1** Synergistic Learning

---

1: **Given**: model set $H_{\text{SynLearn}}$, data providers $\mathcal{D}$.
2: **if** *Feature Pre-Training* **then**
3:    **repeat**
4:       $L_{decom}(\mathcal{Z}_d) \leftarrow \texttt{SQUERY}(\mathcal{Z}_d), d = 1, \ldots, D; \texttt{UPDATE}(g_\mu; \sum_{d=1}^{D} L_{decom}(\mathcal{Z}_d));$
5:    **until** reach end.
6: **end if**
7: **repeat**
8:    *Task & Model Sampling*: $\mu_1, \mu_2 \ldots, \mu_t \sim \mathcal{T}, h_1, h_2 \ldots, h_m \sim \mathcal{M};$
9:    *Reusability Validation*: $g(\mu_i, \mathcal{Z}_d, h_k) \leftarrow \texttt{SQUERY}(\mathcal{Z}_d),$
      $v^*(\mu_i, h_k) \leftarrow \texttt{COMBINE}[\{\psi(\mu_i, \mathcal{Z}_d, h_k)\}_{d=1}^{|\bar{\mathcal{Z}}_{\mu_i}|}], i \in [t], k \in [m], d \in [\bar{\mathcal{Z}}_{\mu_i}];$
10:   *Target Dataset Generation*: $S_{\mu_i} \leftarrow \texttt{MQUERY}(\bar{\mathcal{Z}}_{\mu_i}), i = 1, \ldots, t;$
11:   $\texttt{UPDATE}(\hat{v}; \{S_{\mu_i}\}_{i=1}^{t}, \{h_k\}_{k=1}^{m}, \{v^*(\mu_i, h_k)\}_{i,k=1}^{t,m})$ to minimize Equation 2;
12: **until** reach end.

---

# 4 Learning Method

In this section, we introduce the synergistic learning algorithm and the MRE model.

## 4.1 Synergistic Learning Algorithm

The realization of synergistic learning is illustrated in Algorithm 1. In this section, besides the metric learning step of minimizing Equation 2 (Line 11), we discuss other crucial steps below in general. We consider two general synergistic learning settings. One is named *isolated closure setting*, which indicates that any task closure includes a single data provider. Otherwise, we name it *grouped closure setting*, which indicates that there exist multi-data-provider task closures.

**Task and model sampling (Line 8).** To generate the training data for synergistic learning, tasks and models are needed be sampled. At this stage, no interaction to the data providers is needed.

**Reusability validation (Line 9).** The objective for reusability validation is to acquire value of $v^*$ between any pairs of task and model that are sampled. In general, this can be done by sending models to the data providers and estimate $v^*$ using $v_{valid}$. Under the isolated closure setting, only $\texttt{SQUERY}$ is needed obviously. We could ask the data providers to use sufficiently large data, making $v_{valid}$ an accurate estimator. But the key challenge appears under the grouped closure setting since the statistics used for calculating $v_{valid}$ should be obtained from multiple providers at the same time. To achieve this by $\texttt{SQUERY}$, we require $v_{valid}$ to have a special structure. Specifically, for task $\mu$, let $S_\mu(\mathcal{Z}_d)$ be a sample generated from the marginal distribution of $\mu$ over $\mathcal{Z}_d$, a member of its task closure. We require $v_{valid}(\mu, h) = \texttt{COMBINE}[\{\psi(\mu, \mathcal{Z}_d, h)\}_{d=1}^{|\bar{\mathcal{Z}}_\mu|}]$, in which $\psi(\mu, \mathcal{Z}_d, h) = \mathbb{E}_{S_\mu(\mathcal{Z}_d) \sim \mu(\mathcal{Z}_d)}[\psi(S_\mu(\mathcal{Z}_d), h)]$. In this construction, $\psi(S_\mu(\mathcal{Z}_d), h)$ is a function over single data providers and $\texttt{COMBINE}$ is an aggregation function decided by specific MRE function, thus only $\texttt{SQUERY}$ is needed for calculation.

**Target dataset generation (Line 10).** For isolated closure setting, this step can be done by sampling from a data provider using $\texttt{SQUERY}$. However, for grouped closure setting, $\texttt{MQUERY}$ is necessary for this step since the target datasets must include the raw data. This issue can be relieved by the insight brought from Theorem 3.1: For each task, the number of the target data needed is closely related to the data variance. A feature pre-training stage can be introduced to reduce using $\texttt{MQUERY}$.

**Feature pre-training (Line 2-6).** Feature pre-training is the preparation stage for synergistic learning. It is participated by all data providers, aiming at learning a feature extractor $g_\mu$ which could output low-variance features for the raw data. Once $g_\mu$ is learned, it would be integrated into the MRE model. As a result, in synergstic learning, the data providers only need to use much fewer data for answering $\texttt{MQUERY}$. On the other hand, the feature pre-training stage itself should be done by using only $\texttt{SQUERY}$. We argue that this can be realized by using learning objectives which are *decomposable w.r.t. the data providers*. More specifically, a decomposable objective is the summation of individual objectives, such that each individual objective $L_{decom}^d$ only takes the data from a single

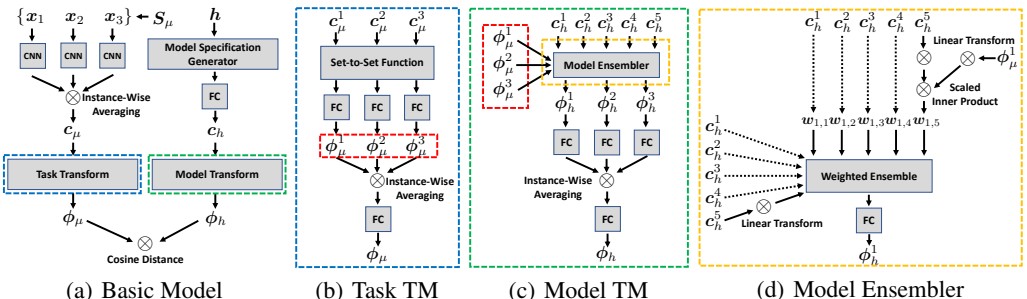

(a) Basic Model      (b) Task TM      (c) Model TM      (d) Model Ensembler

Figure 2: The models proposed for synergistic learning. (a) illustrates the basic model. (b) and (c) illustrate the task and model transform modules (TMs) used in classification. (d) illustrates the model ensembler. The components within the dashed box of the same color are the same.

data provider $\mathcal{Z}_d$ as its input. Then the purpose of the queries reduces to calculating each $L^d_{decom}$, which can be done by SQUERY as expected.

Due to space limitation, we provide more details on how to implement the above synergistic learning steps to different learning scenarios in Section B.

## 4.2  MRE Model

First, we introduce the basic structure of the MRE model defined in Section 3, which is illustrated in (a) of Figure 2. In this model, the target data is first processed by the task feature backbone to generate the feature $c_\mu$. Afterwards, a task transform module is responsible to generate the task representation $\phi_\mu$ which aggregates all the information from the target dataset. For isolated closure setting, the task transform module can be simply an instance-wise averaging operation. In correspondence, before the model feature backbone, a model specification generator is introduced which can be any function that transforms a model into a vector. The model feature $c_h$ is then transformed into the model representation $\phi_h$ using the model transform module. For isolated closure setting, the model transform module can usually be omitted, i.e. $c_h = \phi_h$.

For many learning scenarios, using more elaborate MRE model structure is useful, such as classification. Below we consider small-data transfer learning for classification with supervised pre-trained models. A target task $\mu$ has $L_T$ classes $\{y_l\}^{L_T}_{l=1}$ to distinguish, in which all the classes belong to the task class universe $\mathcal{Y}_T$. Meanwhile, for a pre-trained $L_M$-way classification model $h$, we denote by $\{\hat{y}_{l'}\}^{L_M}_{l'=1}$ the classes that the models are trained on. We assume that all the model classes belong to the model class universe $\mathcal{Y}_M$, which could be different from $\mathcal{Y}_T$. Similar to [Tran et al., 2019, Nguyen et al., 2020, You et al., 2021], we assume that any transfer strategy for this scenario, such as global fine-tuning and head re-training, can be used. While according to our experience, head re-training, in which the feature backbones of the pre-trained models are frozen and only the prediction heads are re-trained, is more proper than global fine-tuning for small-data transfer learning.

The overall structure of the MRE model for classification follows the basic model, but the task and model transform modules have more complex structures. For the task transform module ((b) of Figure 2), we take care of modeling the relationship among task classes. On one hand, instead of directly generating the feature representation of the whole task $\phi_\mu$, the module generates the features $\{\phi^l_\mu\}^{L_T}_{l=1}$ for all task classes first, and then aggregate them to form $\phi_\mu$. Any set-to-set transform module, such as Deep Sets [Zaheer et al., 2017] and Transformer [Vaswani et al., 2017, Ye et al., 2020], can be used. What is essential here is to generate the feature for one task class based on the context information from all other task classes.

More importantly, we propose an attention-based model transform module. This module is motivated by our observations during studying MRE: there is a close connection between $p(y_l, \hat{y}_{l'})$ and model reusability for classification, which is also pointed out by recent studies [Tran et al., 2019, Nguyen et al., 2020]. The module is illustrated in (c) of Figure 2. First, the module takes both the model and task class features $\{c^{l'}_h\}^{L_M}_{l'=1}, \{c^l_\mu\}^{L_T}_{l=1}$ as its inputs and transform them into the *model-attention-aware task class features* $\{\phi^l_h\}^{L_T}_{l=1}$. Subsequently, $\{\phi^l_h\}^{L_T}_{l=1}$ are aggregated to form the

model feature $\phi_h$. The core component of this module is the model ensembler ((d) of Figure 2). For a pair of model and task classes $y_l, \hat{y}_{l'}$, the model ensembler calculates the attention weight of $y_l$ to $\hat{y}_{l'}$, which is $w_{l,l'}$, with the inner-product attention. And then, the model class features are linearly combined by the attention weights to form $\phi_h^l$ which represents the selected model class information on $y_l$. We require $w_{l,l'}$ to be closely related to $p(y_l, \hat{y}_{l'})$ to make it represent the correlation between $y_l$ and $\hat{y}_{l'}$. Therefore, besides the metric loss defined in Equation 2, we introduce an additional *attention supervision loss* to supervise the learning of attention weights. For $w_{l,l'}$, we treat it as the output probability of a binary classifier. The training labels are generated from $p(y_l, \hat{y}_{l'})$. To be specific, we set the label to be one if $p(y_l|\hat{y}_{l'}) > \gamma_1, p(\hat{y}_{l'}|y_l) > \gamma_2$ and zero otherwise, in which $\gamma_1, \gamma_2$ are two thresholds. And then, the attention supervision loss is calculated from the logistic loss. Finally, the overall training loss is formulated as

$$L_{all} = L_{metric} + w_{att}L_{att}, \tag{3}$$

in which $L_{metric}$ is the small-batch version of the metric loss defined in Equation 2, $L_{att}$ is the attention supervision loss defined above, and $w_{att}$ is the weight for $L_{att}$.

## 5 Experiments

In the experiments, we first do metric visualization to verify whether synergistic learning can learn meaningful metric space. Furthermore, we conduct experiments for both in-dataset and cross-dataset MRE to verify the performance of synergistic learning. All experiments are conducted on servers with NVIDIA Tesla V100 GPUs. The code[4] is implemented with TensorFlow [Abadi et al., 2016] (Apache 2.0 License). More details of the experimental setups are discussed in Section C and more experimental results are included in Section D.

### 5.1 Metric Visualization

We adopt two ten-class datasets MNIST [LeCun et al., 1998] and CIFAR-10 [Krizhevsky, 2009] to visualize the task-model metric learned by synergistic learning. For each dataset, we randomly generated 20 five-class pre-trained models, as well as 20 data providers with the same class assignments, for synergistic learning, and another 20 models for testing. We treat each model as five detectors for the corresponding classes and try to use synergistic learning to obtain the detector-data metric. We consider two settings of synergistic learning. The first is learning with full data: all training sets are used for metric learning and there is no feature pre-training stage. The second is learning with part data: 10% of the training set for each class is used for metric learning, and there is a feature pre-training stage involved using the full training set. Figure 3 illustrates the t-SNE [Maaten and Hinton, 2008] visualizations of the learned metric, which is calculated from the testing models and instances. We can see that meaningful metric distance spaces are learned for both learning with full and part data: synergistic learning makes the instances and detectors with the same classes closer. These results provide preliminary proofs for the effectiveness of synergistic learning.

### 5.2 In-Dataset MRE

Next, we use the dSprites dataset [Matthey et al., 2017] for testing in-dataset MRE. dSprites consists of images generated from six latent factors. We select three factors, shape, scale, orientation, to form the task domains. Another two real-valued factors, position X and Y, are used as the prediction targets. There are 720 domains and 1024 instances under each domain in total. For each domain, we randomly select 800 instances as the training set which are used to obtain 720 pre-trained models. The remaining 224 instances are used as the testing set. We randomly select 503 domains as the in-distribution domains and the remaining 217 domains as the out-distribution domains. Under this setting, we treat each in-distribution domain as a data provider, thus there are 503 data providers in total. For synergistic learning, we only use the training sets and models from the in-distribution domains. $v_{valid}(\mu, h)$ is set according to the mean squared error (MSE). We set training $K = 10$, but testing $K = 1$ for verifying the performance under the extremely challenging situation. During testing, we generate testing tasks using the testing sets from either the in-distribution domains or the out-distribution domains. For in-distribution domains, the models used for testing consist of only

---

[4]The code is available on https://github.com/candytalking/SynLearn.

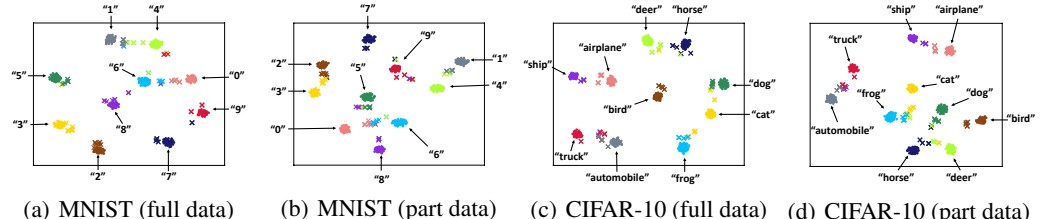

| (a) MNIST (full data) | (b) MNIST (part data) | (c) CIFAR-10 (full data) | (d) CIFAR-10 (part data) |

Figure 3: t-SNE visualization of the learned metric on MNIST and CIFAR-10. (a) and (c) show the results learned with full data. (b) and (d) show the results learned with feature pre-training and part data. The class names for the data clusters (dot markers) are annotated. The detection models (cross markers) have the same color to their corresponding classes (better view in color mode).

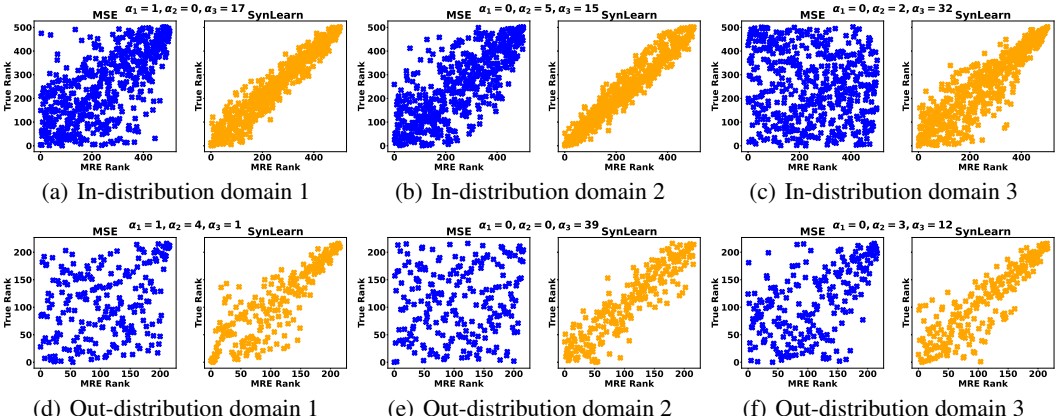

| (a) In-distribution domain 1 | (b) In-distribution domain 2 | (c) In-distribution domain 3 |

| (d) Out-distribution domain 1 | (e) Out-distribution domain 2 | (f) Out-distribution domain 3 |

Figure 4: Performance comparison on dSprites: the MRE score rank vs. the true performance rank for MSE (left subfigure) and synergistic learning (right subfigure). (a-c) show in-distribution results on three random domains, and (d-f) show out-distribution results accordingly. The subfigure titles are the indexes of the domain factors. Please refer to Section D for more results.

in-distribution models. The out-distribution domains follow the similar rule. Figure 4 shows the ranking performance comparison between using MSE directly on the target datasets and synergistic learning (SynLearn). It can be observed that the synergistic learning ourperforms the naive MSE prediction significantly: under the extreme situation where testing $K = 1$, MSE almost fails totally in generating useful rankings, while synergistic learning can still perform desirably.

## 5.3 Cross-Dataset MRE

Finally, we use CIFAR-100 [Krizhevsky, 2009] and MiniImageNet [Vinyals et al., 2016] for cross-dataset MRE experiments. Results on more datasets are included in Section D. We consider two settings for experiments: Reuse CIFAR-100 pre-trained models on MiniImageNet target tasks and the opposite. For each setting, we pre-train 200 20-class models on the source dataset, in which 100 models are used for synergistic learning and the other 100 for testing. We use head re-training as the transfer strategy. For the target dataset, its pre-defined training set is used for synergistic learning and its pre-defined testing set is used for testing. All training and testing tasks are fixed to be five-way classification. For synergistic learning, we randomly generate 100 data providers, each of which holds the data of five target dataset classes. Full training set is used for metric pre-training and 10% of the training set is used for metric learning. For each testing task, 50 instances are sampled from each class to test accuracy. For performance evaluation, we use Kendall's $\tau$-coefficient [ken] to measure the rank correlation between the MRE scores and the testing accuracy. For emphasizing the performance on top-performed models, we also employ the weighted version of Kendall's $\tau$-coefficient, $\tau_w$, for which an exchange between elements with rank $r$ and $s$ (starting from zero) has weight $1/(r+1)+1/(s+1)$ [Vigna, 2015]. We compare synergistic learning (SynLearn) with three state-of-the-art MRE methods: NCE [Tran et al., 2019], LEEP [Nguyen et al., 2020] and LogME

Table 1: Results for the classification experiments. For CIFAR-100 → MiniImageNet, the input shape is $32 \times 32 \times 3$. For MiniImageNet → CIFAR-100, the input shape is $84 \times 84 \times 3$. $\tau$ and $\tau_w$ indicate the Kendall's $\tau$-coefficient and its weighted version calculated from 100 randomly generated five-class target tasks. The results are mean$\pm 95\%$ confidence interval calculated from five random seeds. $K$ indicates the number of training instances per class for each of the testing tasks.

| Setting | Method | $K = 5$ | | $K = 10$ | | $K = 15$ | | $K = 20$ | |
|---|---|---|---|---|---|---|---|---|---|
| | | $\tau$ | $\tau_w$ | $\tau$ | $\tau_w$ | $\tau$ | $\tau_w$ | $\tau$ | $\tau_w$ |
| CIFAR-100 → MiniImageNet | LogME | 0.139±0.070 | 0.180±0.187 | 0.200±0.079 | 0.359±0.231 | 0.243±0.056 | 0.420±0.217 | 0.255±0.042 | 0.418±0.270 |
| | NCE | 0.197±0.020 | 0.393±0.129 | 0.300±0.015 | 0.568±0.097 | 0.365±0.015 | 0.645±0.076 | 0.397±0.008 | 0.661±0.057 |
| | LEEP | 0.282±0.032 | 0.532±0.096 | 0.367±0.016 | 0.649±0.064 | 0.417±0.017 | 0.700±0.051 | 0.441±0.011 | 0.703±0.046 |
| | SynLearn | **0.459±0.022** | **0.714±0.032** | **0.482±0.018** | **0.735±0.016** | **0.497±0.018** | **0.740±0.016** | **0.502±0.011** | **0.750±0.021** |
| MiniImageNet → CIFAR-100 | LogME | 0.166±0.044 | 0.312±0.176 | 0.244±0.083 | 0.413±0.164 | 0.288±0.087 | 0.478±0.163 | 0.310±0.078 | 0.479±0.189 |
| | NCE | 0.173±0.009 | 0.382±0.105 | 0.257±0.020 | 0.496±0.054 | 0.306±0.014 | 0.556±0.081 | 0.345±0.017 | 0.585±0.076 |
| | LEEP | 0.235±0.009 | 0.471±0.065 | 0.323±0.023 | 0.559±0.066 | 0.362±0.019 | 0.619±0.063 | 0.394±0.022 | 0.631±0.076 |
| | SynLearn | **0.419±0.033** | **0.620±0.102** | **0.426±0.027** | **0.642±0.102** | **0.428±0.019** | **0.638±0.076** | **0.431±0.022** | **0.639±0.073** |

[You et al., 2021]. The results are illustrated in Table 1. It can be observed that synergistic learning significantly outperforms other methods for small-data MRE. Note that for results in Table 1, we fix $K = 10$ during synergistic learning, while we observe significantly more robust performance of SynLearn over other approaches when the testing $K$ varies. We conduct ablation studies to verify the effectiveness of the attention supervision, the feature pre-training, and the choice of the training $K$. The results are provided in Section D.

## 6 Related Work

**Direct approaches for MRE.** The direct approaches are based on the statistics calculated on the target training datasets as the MRE scores. The direct approaches involve only simple statistics calculation, meanwhile no auxiliary information is used, thus are usually quite simple and efficient. The representative studies of the direct approaches are [Tran et al., 2019, Nguyen et al., 2020]. In [Tran et al., 2019], the negative conditional entropy (NCE) score is proposed, which is an information-theoretic quantity measuring the entropy for $p(y|\hat{y})$. In [Nguyen et al., 2020], the log expected empirical prediction (LEEP) score is proposed, which can be regarded as an improvement of NCE. The LEEP score also closely related to $p(y|\hat{y})$. But it uses the *soft* prediction probability in calculation, to take the place of the *hard* label assignment calculation in NCE. Thus LEEP uses more information of prediction uncertainty. However, as shown in our experiments, both NCE and LEEP suffer from significant performance degeneration for small-data MRE. This is not surprising since the statistics calculated from small data usually have higher variance.

**Learning approaches for MRE.** The learning approaches conducts learning for MRE. Similar to the existing specification-based approaches, the testing-stage models and tasks are used for learning. In comparison, no testing tasks and models are necessary for synergistic learning. The representative approaches are [You et al., 2021, Achille et al., 2019]. In [You et al., 2021], the logarithm of maximum evidence (LogME) approach is proposed. Different from NCE and LEEP which aim at finding the correlation between the source model predictions and the target outputs, LogME builds the correlation between source model features and the target outputs. Training using target data is necessary for LogME. The advantage of LogME is its wide applicability for different learning problems. While its performance degenerates more significantly than NCE and LEEP under the small-data scenario. This is likely caused by the necessity of learning on the target data which would lead to stronger over-fitting. In [Achille et al., 2019], the Model2Vec approach is proposed. Model2Vec uses metric learning to build task-model metric, which is similar to synergistic learning. But Model2Vec focuses on generating the metric distances for a fixed set of models, thus is a learning approach for MRE. In comparison, synergistic learning generates the task-model metric for future MRE problems in which the models are unseen during synergistic learning.

**Few-shot learning.** In few-shot learning (FSL) [Vinyals et al., 2016, Ha et al., 2017, Snell et al., 2017, Finn et al., 2017], a meta-model is learned to solve future small-data learning tasks. Usually, FSL does not assume to use pre-trained models except for a few cases [Chowdhury et al., 2021]. For MRE, its relationship to FSL is similar to that to transfer learning: MRE can serve as a good preparation step for any FSL task that is allowed to use a pool of pre-trained models.

# 7 Limitations and Future Work

In this work, we proposed the synergistic learning approach for the small-data model reusability evaluation (MRE) problem. In this section, we discuss the limitations of synergistic learning, as well as possible future research directions.

**The task-independent assumption**. Synergistic learning is based on the assumption that the model transferability has little relationship with the task-independent properties of the small sample. But in real problems, *bad samples* indeed exist whose task-independent properties affect transfer performance. We believe that fundamental limits exist for dealing with these samples, thus the relaxation of this assumption is challenging. Note that Theorem 5.1 does not rely on this assumption, thus synergistic learning can be done even when it is not held. While lacking this special property could possibly degenerate the performance. Exploring how to tackle this challenge would be an important future research topic.

**Effectiveness on more learning scenarios**. Due to the resource limitation, we only conduct experiments for supervised pre-trained models. In recent years, reusing unsupervised pre-trained models has been increasingly popular. MRE for unsupervised pre-trained models is a very meaningful topic, on which the research is still missing as far as we know. Note that synergistic learning does not restrict the type of pre-trained models: it can be used whenever a ground-truth MRE function is defined. Thus it can also work for unsupervised pre-trained models in principle. We plan to verify this point in our future researches. Furthermore, in this paper, we only use image datasets for experiments. Applying synergistic learning on application in other modalities is also interesting.

**MRE in learnware**. Learnware [Zhou and Tan, 2022] is a growing research topic about reusing pre-trained models accommodated in a learnware market, which holds various machine learning models submitted by developers all over the world, to enable future user, who knows nothing about the models in the learnware market in advance, no need to build their own machine learning models from scratch. One of the key ingredient of learnware is *model specification*, which enable the models to be efficiently and adequately identified and reused, given the constraint that neither the training data of model developers nor that of users are leaked to the market. Once a set of potentially helpful learnwares have been identified for user, there can be various ways to reuse them to help address users own task. This paper can be viewed as providing a new way to reuse the identified learnwares.

## Acknowledgements

This research is supported by NSFC (61921006, 62206245) and the Collaborative Innovation Center of Novel Software Technology and Industrialization. We would like to thank the anonymous reviewers for their contructive suggestions, as well as Han-Jia Ye, Ming Pang, and Lu Wang for helpful discussions and their thoughtful comments.

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
