# OpenReview forum: "Pre-Trained Model Reusability Evaluation for Small-Data Transfer Learning"
_NeurIPS.cc/2022/Conference — NeurIPS 2022 Accept_

### Official Review · Reviewer_kp6n · 2022-07-11

**Rating:** 6
**Confidence:** 2
**Soundness:** 2 fair
**Presentation:** 2 fair
**Contribution:** 3 good

**Summary:**

This paper focuses on the reusability of pre-trained models with many source data. In particular, this paper aims to predict the transfer learning performance in advance when the target dataset is small. To evaluate model reusability, this paper proposes synergistic learning (metric distance converted to MRE score) as one of the metric learning methods. The proposed method experimentally shows that it is a more useful method to predict the transfer learning performance than baseline methods.

**Questions:**

1. This reviewer intuitively believes that the proposed method is to perform metric learning, which consumes quite a computational cost. Isn't it more computationally expensive than the transfer learning method (fine-tuning the head of networks) performed in this paper?

2. The authors assume that the GT of model transferability depends only on the target task $u$ but does not depend on the random draw of $S_u$ from $u$ but this is not proven theoretically. This reviewer thinks this assumption is hard to realize typically.

Regarding the above questions, please correct me if I am wrong or misunderstood.

**Limitations:**

This reviewer thinks that the authors have properly stated the limitation of this paper.

**Strengths And Weaknesses:**

Strength

The proposed method can robustly calculate MRE scores even in situations where there is little learning data. The proposed method is also a generic method applicable to most problems.

Clarity

From the reader's point of view, the notation of this paper is somewhat confusing because it is different from that of other general transfer learning papers. It is also unclear what circumstances SQUERY and MQUERY assumed. Since there is a lot of space left in Figure 1, it is recommended to explain the situation of SQUERY and MQUERY in pictures.
Finally, some grammatical errors are seen. Therefore, this reviewer recommends that you organize your thesis well.

---

> ### Author Response · Authors · 2022-08-02
> **Thanks very much for the insightful comments.**
>
> Q1: "Since there is a lot of space left in Figure 1, it is recommended to explain the situation of SQUERY and MQUERY in pictures."
>
> A1: We have included the figure in Fig. 1.
>
> Q2: "Isn't it more computationally expensive than the transfer learning method (fine-tuning the head of networks) performed in this paper?"
>
> A2: In our setting, small-data MRE consists of two stages (Fig. 1). Synergistic learning (i.e. metric learning) is at the first stage, which is to obtain the MRE model. Once the MRE model is learned, it will be used in any target MRE tasks at the second stage with low cost. The computational cost is not comparable to existing studies since all of them only consider the second stage. We argue that for small-data MRE, the first stage is crucial, since it can provide essential prior information for the second stage. This is also verified by our experimental results.
>
> Q3: "The authors assume that the GT of model transferability depends only on the target task $\mu$ but does not depend on the random draw of $S_\mu$ from $\mu$ but this is not proven theoretically. This reviewer thinks this assumption is hard to realize typically."
>
> A3: Exploiting the task-model relationship other than the sample-model relationship is essential for small-data MRE. If we don't assume that the samples from the same task are similar, then we cannot expect any prior information obtained from known samples can be applied on new ones. Our assumption is the reflection of this intuition. As discussed in Section 7, in real problems, samples that don't satisfy this assumption may indeed exist. We note that our theory does not rely on this assumption, while the performance of the algorithm may degenerate when the assumption does not hold. We regard this issue as an important future research direction. We have include these discussions in the revision (Line 64-70, 341-345).

---

> > ### Comment · Reviewer_kp6n · 2022-08-08
> > **Thank you for the responses.**
> >
> > Thank you for the responses.
> >
> > There are very helpful to gain a better understanding of the proposed methods.
> >
> > Although it has not been theoretically proven whether the assumption is valid, this reviewer believes that this problem can be achieved in subsequent studies.
> >
> > The authors have almost cleared my question so this reviewer is willing to improve my score.

---

### Official Review · Reviewer_xY6u · 2022-07-11

**Rating:** 6
**Confidence:** 3
**Soundness:** 3 good
**Presentation:** 3 good
**Contribution:** 3 good

**Summary:**

This paper proposes synergistic learning, a new metric for Model Reusability Evaluation (MRE) of pre-trained models to downstream tasks with small data. This task-model metric is learned using a set of pre-trained models and one or more data providers. Given a validation MRE function, synergistic learning establishes a metric between data providers and pre-trained models so that this metric distance can be used as an MRE score. The authors propose four different types of synergistic models: 1) basic model, 2) task transform model, 3) model transform model, 4) ensembler model. Experiments are run on benchmark MNIST, CIFAR-10, dSprites, CIFAR-100 and MiniImagenet datasets, and show that the proposed method is competitive as a MRE metric compared to SOTA methods.

**Questions:**

Q1: In the experiments, can you clarify how many data providers are you actually using?

Q2: How do the experimental results of the proposed MRE model relate to the bounds presented in Theorem 3.1, especially in terms of the number of the training datasets required when the feature variance is small?


**Limitations:**

Limitations are clearly stated.

**Strengths And Weaknesses:**

Strengths:
- The paper tackles the important problem of learning to select pre-trained models for downstream tasks with small training data.
- The paper is technically sound and the claims are accompanied by theoretical and experimental analyses.
- The experiments show that the proposed method is successful as a MRE metric, compared to counterpars.

Weaknesses:
- The main weakness that I see is related to clarity and organization of the paper, especially in terms of definitions and assumptions. For example, it seems that feature extractors are an important part of the proposed method, however this assumption is not clearly stated until later in the paper. Also, important assumptions for the small data transfer scenario such as re-training of heads only (lines 187-195) are provided later in the paper. In general, the presence of definitions and assumptions in several parts of the paper make it difficult to read, and a big effort is required for understanding the ideas, assumptions and limitations of the proposed approach. I would suggest the authors to reorganize the paper in such a way that assumptions and definitions of the general approach are provided first, then an explanation of the proposed method (the theory and the algorithm).

---

> ### Author Response · Authors · 2022-08-02
> **Thanks very much for the constructive comments.**
>
> Q1: "it seems that feature extractors are an important part of the proposed method, however this assumption is not clearly stated until later in the paper."
>
> A1: We have made the introduction of the model structure clearer in the paper (Line 99-104).
>
> Q2: "important assumptions for the small data transfer scenario such as re-training of heads only (lines 187-195) are provided later in the paper"
>
> A2: In fact, our method does not restrict to use head re-training. This is just the choice in the experiments. We have made this clarified in the paper (Line 59-60, 197-201, 274-275).
>
> Q3: "I would suggest the authors to reorganize the paper"
>
> A3: To improve paper organization, we have added the notification that Section 3 can be skipped for readers who are interested more on the algorithm than the theory (Line 106-109). We will further improve the paper organization in the future revision.
>
> Q4: "In the experiments, can you clarify how many data providers are you actually using?"
>
> A4: We have included the explanations in the revision (Line 239, 257-259, 277-278).
>
> Q5: "Q2: How do the experimental results of the proposed MRE model relate to the bounds presented in Theorem 3.1, especially in terms of the number of the training datasets required when the feature variance is small?"
>
> A5: For experiments in 5.1 and 5.3, we include the results for synergistic learning with feature pre-training. They show that by using feature pre-training to reduce feature variance, we only need to use 10% of the training data to do metric learning. We will include ablation study results for changing this ratio in the future revision.

---

> > ### Comment · Reviewer_xY6u · 2022-08-09
> > **Response to authors**
> >
> > I appreciate the effort of the authors on revising the paper and providing a new version.
> >
> > I am keeping my score unchanged as I think that this is a technically solid paper with moderate to high impact, and I have no major concerns.

---

> ### Comment · Area_Chair_zyoX · 2022-08-08
> **Response needed**
>
> Dear Reviewer xY6u,
>
> Please kindly respond to the rebuttal provided by the authors and/or engage in the discussion with them. If it addresses your concerns, please react accordingly. Otherwise, please elaborate in your review on why you think the rebuttal/discussion is inadequate. Thank you.
>
> Best, AC

---

### Official Review · Reviewer_wt4R · 2022-07-12

**Rating:** 5
**Confidence:** 3
**Soundness:** 3 good
**Presentation:** 2 fair
**Contribution:** 3 good

**Summary:**

This work investigates the estimation of model reusability when the size of target data is small. Specifically, features are extracted for tasks and pre-trained models, respectively. Then, the metric distance between them is optimized as MRE score. The theoretical analysis and the empirical study confirm the effectiveness of the proposed method.

**Questions:**

My major concern is about the application for unsupervised pre-trained models, which demonstrate better transfer performance than supervised counterparts.

**Limitations:**

Yes.

**Strengths And Weaknesses:**

Strong.
1.	With the development of pre-training, evaluating the reusability of existing pre-trained models appropriately is important for applications.
2.	The framework is elaborated with the example of classification.
3.	The empirical study shows that the proposed method can learn better MRE score than existing methods.

Weak.
1.	Some notations are not clear. For example, the loss in Line 108 contains a term of $v(\mu’, h’)$ without any explanation about how to obtain it. Besides, it is better to elaborate the feature generator for models, i.e., $c_h$ and the validation MRE function $v_{valid}$ with examples to help illustrate.
2.	For classification, the proposed method requires the label space for pre-trained models which is unavailable for models learned without labels. It may limit the application of the proposed method, which is only applicable for supervised pre-trained models.
3.	The current experiments only contain models pre-trained on target data. It is better to include existing pre-trained models as in LogME to demonstrate the effectiveness for a diverse model zoo.

---

> ### Author Response · Authors · 2022-08-02
> **Thanks very much for the detailed comments.**
>
> Q1: "the loss in Line 108 contains a term of $v(\mu', h')$  without any explanation about how to obtain it".
>
> A1: This is a typo. We have fixed it (Line 114-115).
>
> Q2: "it is better to elaborate the feature generator for models, i.e., $c_h$ and the validation MRE function $v_{valid}$ with examples to help illustrate."
>
> A2: Due to the space limitation, we put the examples in the appendix (Section B and Section C). We will try to put them in the main paper in the future revision.
>
> Q3: "It may limit the application of the proposed method, which is only applicable for supervised pre-trained models." “My major concern is about the application for unsupervised pre-trained models, which demonstrate better transfer performance than supervised counterparts.”
>
> A3: In theory, the proposed method can also be applied on unsupervised pre-trained models. While the price to pay is that the model ensembler illustrated in Fig. 2 cannot be used. We need to use simpler model structure. But we still believe that synergistic learning should work for it. We will add the experiments for unsupervised pre-trained models in the future revision.
>
> Q4: "The current experiments only contain models pre-trained on target data."
>
> A4: In the experiments, the models are pre-trained on the source data other than the target data. Please refer to Line 259-260, 273-274 for the details.

---

> > ### Comment · Reviewer_wt4R · 2022-08-08
> > **Final rating**
> >
> > The rebuttal partially addresses my concerns while the application of unsupervised pre-trained model is still challenging. So I will keep my rating.

---

> ### Comment · Area_Chair_zyoX · 2022-08-08
> **Response needed**
>
> Dear Reviewer wt4R,
>
> Please kindly respond to the rebuttal provided by the authors and/or engage in the discussion with them. If it addresses your concerns, please react accordingly. Otherwise, please elaborate in your review on why you think the rebuttal/discussion is inadequate. Thank you.
>
> Best, AC

---

### Author Response · Authors · 2022-08-02
**Rebuttal Revision**

We would like to thank all the reviewers for the constructive comments. The submission has been revised according to the reviewers' suggestions, in which the changes are highlighted with blue color. In special, the major efforts are made to improve paper clarity. Please refer to the individual responses for more details.

---

### Meta-Review · Area_Chair_zyoX · 2022-08-27

**Recommendation:** Accept
**Confidence:** Less certain

**Metareview:**

There is a consensus among the expert reviewers that this paper tackles an important problem, is technically sound, and has a sufficient contribution for publication at NeurIPS2022. Synergistic learning is still in its infancy and as a result, requires visibility from the community. The proposed methods for calculating model reusability evaluation (MRE) metrics in this paper will serve as an important baseline for subsequent research in this direction. Personally, I also appreciate this research direction as it will be a crucial part of the democratization of AI.

The main reservation is the clarity and presentation of the paper. The authors attempted to address them by providing clarification during the discussion phase and by revising the manuscript accordingly, which I really appreciate. The reviewers also acknowledged and responded positively to the authors' responses, either maintaining their high scores or increasing them. Hence, I recommend that the authors take special care of this issue in the camera-ready version.

**Award:**

No

---

### Decision · Program_Chairs · 2022-09-14

Accept